# Enhancing Drought Tolerance in Wheat Cultivars through Nano-ZnO Priming by Improving Leaf Pigments and Antioxidant Activity

Syed Farhat Abbas [1], Muhammad Adnan Bukhari [1], Muhammad Aown Sammar Raza [1], Ghulam Hassan Abbasi [2], Zahoor Ahmad [3,*], Mashael Daghash Alqahtani [4], Khalid F. Almutairi [5], Elsayed Fathi Abd_Allah [5] and Muhammad Aamir Iqbal [6]

[1]  Department of Agronomy, Faculty of Agriculture and Environment, The Islamia University of Bahawalpur, Bahawalpur 63100, Pakistan
[2]  Institute of Agro-Industry and Environment, Faculty of Agriculture and Environment, The Islamia University of Bahawalpur, Bahawalpur 63100, Pakistan
[3]  Constituent College, University of Central Punjab, Yazman Road, Bahawalpur 63100, Pakistan
[4]  Department of Biology, College of Science, Princess Nourah bint Abdulrahman University, P.O. Box 84428, Riyadh 11671, Saudi Arabia
[5]  Plant Production Department, College of Food and Agricultural Sciences, King Saud University, P.O. Box 2460, Riyadh 11451, Saudi Arabia
[6]  Department of Agronomy, Faculty of Agriculture, University of Poonch Rawalakot, Rawalakot 12350, Pakistan
*   Correspondence: zahoorahmadbwp@gmail.com

**Abstract:** Climate change, global warming, stagnant productivity of wheat and food security concerns owing to frequent spells of drought stress (DS) have necessitated finding biologically viable drought-mitigation strategies. A trial was conducted to test two promising wheat cultivars (Ujala-16 and Zincol-16) that were subjected to pre-sowing priming treatments with different doses of ZnO nanoparticles (NPs = 40, 80, 120 and 160 ppm) under 50% and 100% field capacity (FC) conditions. The ZnO NPs were prepared with a co-precipitation method and characterized through X-ray diffraction (XRD) and with a scanning electron microscope (SEM). For comparison purposes, untreated seeds were sown as the control treatment. The response variables included botanical traits (lengths, fresh and dry wrights of root and shoot), chlorophyll (a, b and total) contents, antioxidant and proline contents and nutrients status of wheat cultivars. The results showed that DS significantly decreased all traits of wheat cultivars, while ZnO NPs, especially the 120 ppm dose, remained superior by increasing all botanical traits at 100% FC. In addition, ZnO NPs increased the chlorophyll *a* (1.73 mg/g FW in Ujala-16 and 1.75 mg/g FW in Zincole-16) *b* (0.70 mg/g FW in Ujala-16 and 0.71 mg/g FW in Zincole-16) and total chlorophyll content (2.43 mg/g FW in Ujala-16 and 2.46 mg/g FW in Zincole-16) by improving the activity of antioxidant and proline content. Moreover, plant nutrients such as Ca, Mg, Fe, N, P, K, and Zn contents were increased by ZnO NPs, especially in the Zincol-16 cultivar. To summarize, Zincol-16 remains superior to Ujala-16, while ZnO NPs (120 ppm dose under 100% FC) increases the growth and mineral contents of both wheat varieties. Thus, this combination might be recommended to wheat growers after testing further in-depth evaluation of more doses of ZnO NPs.

**Keywords:** seed priming; morphology; leaf pigments; antioxidant; nutrient analysis; zinc oxide; drought stress; wheat



## 1. Introduction

Globally, wheat (*Triticum aestivum* L.) is an important staple food for millions of people and is being cultivated in all inhabitable continents [1,2]. Due to climate change and the increasing food demand, improving wheat yield per unit area has become pressing [3,4]. It is projected that if the current changes in the climate continue, then wheat production

may decline by over 50% in next two decades [5,6]. To address these challenges, the use of nanotechnology for developing nanoparticles (NPs) to enhance plant growth and yield under stressful conditions hold potential and bright prospects [7,8]. Additionally, the application of NPs for boosting wheat yield through the amelioration of drought's deleterious impacts might play a significant role in addressing the food security concerns of the modern era [6,9,10]. Recently, many NP products such as nano-fertilizers, nano-pesticides, and nano-sensors have already been tested and employed to enhance crops productivity [11,12]. The NPs help to increase the nutrient-use efficiency of plants. Because of their small size, NPs cover a larger surface area and directly improve the physiological functions of crop plants [11]. Various methods have been used to supply NPs to plants such as seed coating, soil application and foliar spray under a stressful environment. The application of NPs through seed priming (SP) is an innovative technique that may improve seed vigor, which is a prerequisite of better stand establishment under normal or even stressful environments [12]. In the SP method, seeds are soaked in water (hydro-priming) and aerated solutions (osmo-priming) for a specific time period that can trigger the metabolic processes (MPs). These MPs are generally activated during the early phase of germination (pre-germinate metabolism) and therefore increase the rate of germination and seedling establishment [13,14]. Different priming solutions (polyethylene glycol, hormones, nutrients, organic salts, etc.) help the plants to off-set the adverse effects of abiotic stresses including DS [15,16]. For the SP, different NPs such as gold (Au), silver (Ag), iron (Fe), copper (Cu), zinc oxide (ZnO), zinc (Zn), carbon-based NPs fullerene and carbon nanotubes have been used to mitigate the harmful effects of DS [15,17].

There are research and knowledge gaps regarding the optimum source and dose of NPs applied as SP (seed priming) agents because different NPs tend to use atypical mechanisms for mitigating the adverse effects of DS. The studies of nano-priming effects on seed germination, early stand establishment and plant growth in wheat under normal and stressful conditions have not been elucidated so far. Zinc (Zn) is an essential micronutrient required for plant development and growth as it is an essential component of over 300 proteins and enzymes synthesized by plants under normal and scant water conditions [16,18]. Zn is needed for a variety of physiological functions such as pollination, growth regulation, antioxidant function and protein synthesis. Moreover, it also plays a vital role in photosynthesis, maintenance of cell membranes, detoxification of free radicals and gene expression under water-deficient conditions [19–21].

The primary challenge of Zn application as ZnO is the low solubility of Zn in soil and higher losses to terrestrial ecosystems [22–24]. However, NPs of ZnO might overcome these problems due to their high solubility, availability and reactivity within plant tissues [25]. Additionally, these have drawn the attention of researchers owing to their unique photo-oxidizing, physiological and biochemical capacity, as well as their unique functions in plants physiological functions [6,26]. The SP with NPs of ZnO significantly increased the Zn content in the primed seeds, leading to improved seedling vigor, growth and economic yield [6,26,27]. Likewise, Tului et al. [28] inferred that ZnO NPs had positive effects on the growth of chickpeas (*Cicer arientinum*), while their growth-promoting impact has also been recorded for mung bean [29], cucumber [30]), alfalfa [31] and tomato [32]. However, previous studies present contrasting findings pertaining to the most superior and effective dose of ZnO NPs for ameliorating the adverse effects of DS, while considerable research gaps also exist regarding their efficiency for different genotypes of wheat. Thus, the research hypothesis of this study is that wheat genotypes might respond differently to varying doses of ZnO NPs under atypical DS levels. Therefore, this study aims to evaluate wheat cultivars response to different doses of ZnO nanoparticles in terms of numerous botanical, physiological and biochemical traits, while the ultimate goal was to identify the drought resistant wheat cultivar and the most superior dose of ZnO NPs for ameliorating the adverse effects of DS under changing climate scenarios.

## 2. Materials and Methods

### 2.1. ZnO Nanoparticle Synthesis

By using the co-precipitation process, the ZnO NPs were prepared by following the suggested procedure [33]. Freshly made NaOH solution was gradually added, drop by drop, to the $ZnSO_4 7H_2O$ solution at a 2:1 ratio. The resulting milky white slurry was swirled on a magnetic stirrer for 12 h. The ZnO precipitates were prepared, then filtered (Whatman No. 42), followed by a thorough deionized water wash thrice. After that, precipitates were dried in a forced air oven at 105 °C. Subsequently, with the help of a pestle and mortar, dried precipitates were crushed before being calcined at 550 °C for two hours. Figure 1 shows a step-by-step procedure of the preparation of the NPs. The balanced reaction equation is as follows:

$$2NaOH + ZnSO_4 \cdot 7H_2O \rightarrow ZnO\ (ppt) + Na_2SO_4 + 8H_2O$$

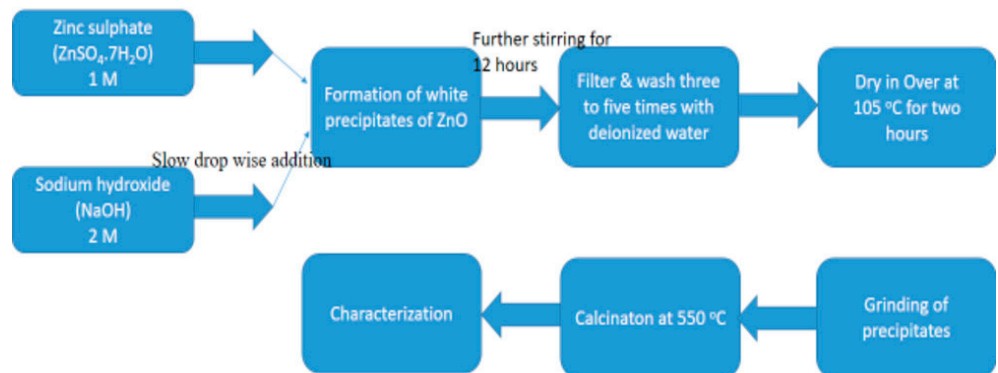

**Figure 1.** A flow chart presentation of ZnO nanoparticles synthesis.

X-ray powder diffraction (XRD), Zeta sizer and scanning electron microscopy (SEM) analysis were used to characterize the prepared nanoparticles [34].

### 2.1.1. Characterization of Synthesized Nanoparticles
X-ray Powder Diffraction (XRD)

A study using an X-ray diffractometer was conducted to ascertain the size and crystalline phase structure of ZnO-NPs. The Debye-Scherrer equation was used to determine the size of the ZnO NPs crystals [35].

$$D = \frac{k\lambda}{\beta \cos\theta}$$

where, $D$ = the mean crystalline size, k = Scherer constant (0.89), $\lambda$ = X-ray wavelength and $\beta$ = full width of half peak maximum (FWHM) intensity (in radians), denoted as $\Delta\ (2\theta)$ and $\theta$ = Bragg's diffraction angle.

### 2.1.2. ZnO Nanoparticles Suspension Formulation

For each nanoparticle application, pre-weighed amounts of the necessary ZnO nanoparticles for wheat were suspended directly in deionized water in a flask, and the particles were then spread out through ultrasonic vibration in a water bath sonicator for 30 min, just before the application of treatment. Sonication was done independently for each replication and treatment.

### 2.2. Pot Experiment

The experiment was conducted at the wire house of Islamia University, Bahawalpur, Pakistan (29.3544° N, 71.6911° E). The planting material included wheat cultivars such as

Zincol-16 (fortified) and Ujala-16 (non-fortified) that were subjected to the pre-sowing SP of ZnO NPs.

For SP, different doses of ZnO NPs (40, 80, 120 and 160 ppm) were prepared, and then seeds of both wheat varieties were primed for 12 h at 25 °C. The unprimed seeds were used as the control. After 12 h of priming, seeds were washed with distilled water thrice (three min for each), and allowed to dry at room temperature. The seed was picked up and added into the bag for preserved and stored at room temperature. Pots were separated into two groups as one group was kept at a well-watered condition (control plants received water applied according to 100% field capacity) while the other group was given withholding water stress (water applied at 50% field capacity) four weeks after the sowing. In each pot, 10 seeds were sown, but after seedling establishment, 5 healthy plants were kept in pots. Fertilizers such as nitrogen (N), phosphorus (P) and potash (K) were used at 120–90–60 kg N, P and K ha$^{-1}$. A whole quantity of potash fertilizer and phosphorus fertilizer and half a dose of N were mixed with soil and then filled with the pots. The remaining N was added at the onset of tillering. No herbicide was used and weeds were removed manually. All of the parameters under investigation were recorded by following the standard protocols.

### 2.3. Agronomic Parameters

The agronomic parameters such as shoot and root lengths, their fresh and dry weights were noted by randomly selecting three plants from each treatment. The roots and shoot were weighed separately using a digital balance and placed into the oven at 80 °C until a constant weight was obtained. The plant height was measured with the help of a meter rod. Similarly, the leaf area per plant was noted using a digital leaf area meter.

### 2.4. Physiological and Biochemical Parameters

The plant leaf (2nd top leaf) was used to determine the leaf water potential from each treatment using slender-type pressure chamber between 8:00 to 10:00 a.m. In case of relative water content, the flag leaf of the wheat plants was detached and weighed using a digital electrical balance (Choy, MK-500C) and then the leaves were dipped in the test tube containing distilled water for 24 h. After 24 h, the leaves were taken out and wiped with tissue papers, and their turgid weight (TW) was noted. Then the leaves were dried at 70 °C for 72 h to record their dry weight (DW). Due to this formula, the relative water content was noted as

$$\text{RWC} = \{\text{FW} - \text{DW} | \text{TW} - \text{DW}\} \times 100 \tag{1}$$

For investigation of chlorophyll *a*, *b* and total chlorophyll, the sample of 100 mg leaves from treated and untreated plants from each treatment was taken and placed in 20 mL of chilled acetone (80% *v/v*). At 663 and 645 nm, the absorption was measured using a spectrophotometer (AA-7000, SHIMADZU). The chlorophyll *a*, *b* and total Chl contents were determined using the following equations [36]:

$$\text{Chlorophyll } a \text{ (mg/g FW)} = 12.7 \, (A_{663}) - 2.69 \, (A_{645})] \times V\} \div (1000 \times W) \tag{2}$$

$$\text{Chlorophyll } b \text{ (mg/g FW)} = 22.9 \, (A_{645}) - 4.68 \, (A_{663})] \times V\} \div (1000 \times W) \tag{3}$$

$$\text{Total chlorophyll (mg/g FW)} = 20.2 \, (A_{645}) + 8.02 \, (A_{663})] \times V\} \div (1000 \times W) \tag{4}$$

For the estimation of total proline content, 0.5 g fresh leaf material was grounded and mixed in 10 mL of 3% sulfosalicylic acid [37], and then filtered through Whatman No. 40 filter paper. The filtrate material of 2 mL was taken in a 25 mL test tube and left for reaction. Then 2 mL of acid ninhydrin solution and 2 mL of glacial acetic acid were mixed and test tubes were heated at 100 °C. After completing all of the reactions, its absorbance was measured at 520 nm through a spectrophotometer (model AA-7000, Shimadzu, Japan).

The proline concentration was calculated from the standard curve and determined on a fresh weight basis as follows:

$$\text{Proline (μmoles g}^{-1}\text{ FW)} = [(\text{μg proline/mL}) \times (\text{mL toluene})/115.5\text{ μg/μmole}]/[(\text{g sample})/5] \tag{5}$$

Antioxidant enzyme activities such as peroxidase (POX), catalase (CAT), ascorbate peroxidase (APX) and glutathione peroxidase were noted through spectrophotometer (AA-7000, SHIMADZU). The collected leaves were homogenized in a medium composed of 50 mM phosphate buffer with 7.0 pH and 1 mM dithiothreitol as described by [31,38].

The ascorbic peroxidase (APX) was measured by observing the decrease in absorbance of ascorbic acid at 290 nm in a reaction mixture of 1 mL, which contained the 50 mM phosphate buffer having pH 7.6, 0.1 mm Na-EDTA, 0.25 mM ascorbic acid and 12 mM $H_2O_2$ sample extract as described by [39]. The activity of the catalases was observed with the conversion rate of hydrogen peroxide into the water molecules and $O_2$, according to the method of Chance and Maehly [40]. This reaction was observed in a 3 mL solution containing 50 mM of phosphate buffer having pH 7, $H_2O_2$ 5.9 mM and 0.1 mL of the extract of the enzyme. The enzyme's catalyzing activity was calculated by observing the decrease in the rate of absorbance at 240 nm after 20 s due to the consumption of $H_2O_2$. The absorbance change at 0.01 units per minute was defined as the 1-unit catalase activity.

The glutathione peroxidase activity was observed to measure the peroxidation and hydrogen peroxide with the guaiacol and its role as an electron donor [40]. The solution of POD in which the reaction occurs consists of 50 mM of phosphate buffer having a pH of 5, 20 Mm of guaiacol, 40 mM of $H_2O_2$ and 0.1 mL of enzyme extract. An increased absorption due to the formation of tetraguaicol at 470 nm was assessed after the 20-s interval. A single unit of enzyme is considered to be the quantity of enzyme which mainly causes the raise in OD value of 0.01 in 1 min. This enzyme's activity was shown as units $\text{min}^{-1}\text{ g}^{-1}$. The superoxide dismutase (SOD) activity was observed by noting down the inhibition of the photo-reduction of nitro blue tetrazolium (NBT) by the SOD enzyme. The reaction mixture contained the 50 mM sodium phosphate buffer having pH 7.6, 0.1 mM EDTA, 12 mM L-methionine, 10 μM riboflavin, 50 μM NBT, 50 mM sodium carbonate and 100 mL of crude extract in 3 mL final volume. A control reaction was performed without the extract. The SOD reaction was carried out by exposing the reaction mixture to white light at room temperature for 15 min. After 15 min of incubation, the absorbance was measured at 560 nm through the spectrophotometer. A single unit of SOD was described as the concentration of an enzyme that caused a 50% photochemical reduction of NBT.

### 2.5. Plant Nutrient Analysis

For the plant nutrient analysis, first, 0.5 g of dried ground material was added in digestion tubes, and 5 mL of concentrated $H_2SO_4$ was maintained in each test tube (Wolf, 1982). All of the tubes were placed in an incubator at room temperature and 0.5 mL of $H_2O_2$ (35%) was poured down on the sides of the digestion tube, and continuously heated at 35 °C temperature for 30 min. The tubes then remained in the digestion block heat until fumes were produced. After heating, the tube was removed from the block and stayed for the cooling process. Then, 0.5 mL of $H_2O_2$ was added slowly and the tube was placed again into the digestion block. These steps were repeated till the material was cooled and became transparent. In the volumetric flask, the extract was maintained at 50 mL. Thereafter, the filtration process was performed to determine the Ca, Mg, Zn, P, D, Fe and Zn contents through a flame atomic absorption spectrophotometer (model AA-7000, SHIMADZU). Moreover, nitrogen was measured through the Kjeldahl method, which indirectly estimates N content in the plants tissues [41].

### 2.6. Statistical Analysis

The collected data were analyzed by employing one-way analysis of variance (ANOVA) technique using the statistical software "Statistix" (version 8.1, Analytical Software, Talla-

hassee, FL, USA). Thereafter, a least significant difference (LSD) test was employed at the *p* value ≤ 0.05 to determine the significance among treatment means [42].

## 3. Results

The results show that DS reduces the morphological parameters such as the shoot and root lengths of both wheat varieties, however, SP with ZnO NPs significantly enhances the morphological parameters compared with untreated seeds. Seed priming with ZnO NPs significantly increases the shoot and root lengths (Table 1) as the maximum shoot (69.34 cm) and root length (23.56 cm) are measured for Zincol-16 when the ZnO NPs (120 ppm) is applied under well-watered (100% FC) conditions. Similarly, seeds of Ujala-16 primed with 120 ppm of ZnO NPs and grown under 100% FC show higher shoot (63.20 cm) and root length (21.47 cm) as compared with 160 ppm of ZnO NPs (58.95 cm shoot length and 20.04 cm root length). Zincol-16 is more responsive to ZnO NPs as compared with Ujala-16 under well-watered (100% FC) and water-stressed (50% FC) conditions. Wheat plants from untreated seeds record the minimum shoot (24.36 to 29.89 cm) and root length (8.27 to 10.15 cm in Ujala-16 and Zincole-16, respectively). Similarly, shoot and root fresh and dry weights significantly increase with seed priming with ZnO NPs. All of the ZnO NPs treatments increase shoot and root dry weights as compared with untreated seeds. The maximum shoot fresh (13.26 g) and dry weight (5.45 g) is recorded in Zincol-16 when primed with ZnO NPs at 120 ppm and grown at 100% FC, followed by the same wheat variety at 160 ppm and Ujala-16 primed with 120 ppm of ZnO NPs. The DS decreases the shoot fresh and dry weights but the maximum reduction in shoot fresh (4.66 g) and root dry weight (1.91 g) are recorded for Ujala-16 grown without seed treatment.

**Table 1.** Effect of ZnO NPs on shoot, root length, and shoot fresh and dry weight of wheat plants.

| Wheat Varieties | Drought Stress | ZnO Levels | Shoot Length (cm) | Root Length (cm) | Shoot Fresh Weight (g) | Shoot Dry Weight (g) |
|---|---|---|---|---|---|---|
| Ujala-16 | 50% FC | Control | 24.36 j | 8.27 j | 4.66 j | 1.91 j |
| | | 40 ppm | 35.74 h | 12.14 h | 6.83 h | 2.80 h |
| | | 80 ppm | 42.21 g | 14.34 g | 8.07 g | 3.31 g |
| | | 120 ppm | 52.77 e | 17.93 e | 10.09 e | 4.14 e |
| | | 160 ppm | 47.97 f | 16.30 f | 9.17 f | 3.77 f |
| | 100% FC | Control | 40.69 g | 13.83 g | 7.78 g | 3.19 g |
| | | 40 ppm | 53.78 de | 18.27 de | 10.29 de | 4.22 de |
| | | 80 ppm | 54.67 de | 18.57 de | 10.46 de | 4.29 de |
| | | 120 ppm | 63.20 b | 21.47 b | 12.09 b | 4.96 b |
| | | 160 ppm | 58.98 c | 20.04 c | 11.28 c | 4.63 c |
| Zincole-16 | 50% FC | Control | 29.89 i | 10.15 i | 5.72 i | 2.35 i |
| | | 40 ppm | 40.39 g | 13.73 g | 7.73 g | 3.17 g |
| | | 80 ppm | 47.02 f | 15.98 f | 8.99 f | 3.69 f |
| | | 120 ppm | 55.74 d | 18.94 d | 10.66 d | 4.38 d |
| | | 160 ppm | 54.15 de | 18.40 de | 10.36 de | 4.25 de |
| | 100% FC | Control | 46.88 f | 15.93 f | 8.97 f | 3.68 f |
| | | 40 ppm | 52.93 e | 17.99 e | 10.12 e | 4.16 e |
| | | 80 ppm | 58.84 c | 19.99 c | 11.26 c | 4.62 c |
| | | 120 ppm | 69.34 a | 23.56 a | 13.26 a | 5.45 a |
| | | 160 ppm | 64.05 b | 21.76 b | 12.25 b | 5.03 b |
| LSD value at 5% probability level | | | 2.25 | 0.76 | 0.43 | 0.17 |

Fc = Field capacity. Values having dissimilar letters within same column indicate significant difference at 5% probability.

Priming of wheat seed with ZnO NPs results in significantly higher leaf area (LA) per plants as compared with untreated seeds (Table 2). The Zincol-16 cultivar exhibits the maximum LA (417 cm$^2$) when seeds are primed with ZnO NPs (120 ppm) and sown under 100% FC as compared to Ujala-16 (318 cm$^2$). The leaf water potential represents a useful index of soil water stress and provide insights regarding plant-water relationships. Data shows that wheat plants under DS show a higher leaf water potential. Plants grown from untreated seeds show higher water potential. In plant leaf water potential, the improvement is shown after the treatment of ZnO NPs, and minimum leaf water potential ($-24.00$ bars) is recorded for Ujala-16 grown subjected to SP with ZnO NPs (160 ppm) under 100% FC, followed by Zincol-16 ($-31.66$ bars). Likewise, the relative water content (RWC) increases with SP with ZnO NPs (Table 2) as the maximum RWC is observed for Zincol-16 (82.66%) in response to ZnO NPs applied at the rate of 120 ppm under 100% FC, which is on par with Ujala-16 treated seeds with ZnO NPs at 160 ppm (82.00%). Both wheat varieties show minimum RWC when seeds are not primed and grown in water stress conditions (50% FC), however, the lowest RWC (62.66%) is noted for Zincol-16. Wheat plants show a higher content of Chl a, b and total Chl content when seeds are primed with ZnO NPs. All the treatments of ZnO NPs improve Chl *a*, *b* and total Chl over the plants grown from untreated seeds (Table 3). The DS (50% FC) reduce the Chl *a*, *b* and total Chl as compared with well-watered conditions (100% FC). The maximum Chl *a* (1.69 mg/g FW), *b* (0.71 mg/g FW) and total Chl (2.46 mg/g FW) contents are recorded for Zincol-16 under ZnO NPs applied at 120 ppm with 100% FC, which is at par with Ujala-16 primed with ZnO NPs at 160 ppm. The lowest Chl *a* (1.33 mg/g FW), *b* (0.43 mg/g FW), and total Chl (1.76 mg/g FW) content is recorded for Zincol-16 when seeds are sown under drought-stressed conditions without ZnO treatment followed by Ujala-16.

Interestingly, the untreated seeds of Ujala-16 under DS conditions show higher proline content. ZnO NPs also significantly increase the proline content of wheat. The maximum proline content (5.13 µmoles g$^{-1}$ FW) is recorded for Zincol-16 when seeds are primed with ZnO NPs at 120 ppm and grown under 50% FC level, followed by Zincol-16 at 120 ppm (4.74 µmoles g$^{-1}$ FW) and Ujala-16 seed treatment with ZnO NPs at 120 ppm (4.68 µmoles g$^{-1}$ FW) grown under 50% FC.

Moreover, the activity of antioxidants such as catalase (CAT), superoxide dismutase (SOD), glutathione peroxidase (GPx) and ascorbate peroxidase (APX) are increased in plants when grown under DS conditions, while ZnO NPs also increase the antioxidant contents in wheat. However, maximum CAT (446.68 Units m$^{-1}$ g$^{-1}$ FW), SOD (617.08 units m$^{-1}$ g$^{-1}$ FW), APX (2.68 ABA digested g$^{-1}$ FW h$^{-1}$) and GPx (165.66 units m$^{-1}$ g$^{-1}$ FW) activity is recorded for untreated Ujala-16 under DS conditions that are statistically on par with Zincol-16 under the same set of experimental conditions. Seed treatment with ZnO NPs at 40 ppm shows higher antioxidant activity as compared with the higher doses of ZnO NPs (Table 4).

Similarly, DS also affects the nutrient concentration in wheat plants of both cultivars as N, P, K, Ca, Mg, Zn and Fe in plants are significantly reduced (Tables 5 and 6). However, SP with ZnO NPs remain effective in enhancing the uptake of N, K and Zn while having a lesser impact on P, Ca, Mg and Fe. The maximum concentration of P (5.82 g kg$^{-1}$ FW), K (16.31 g kg$^{-1}$ FW), N (7.50 g kg$^{-1}$ FW) Ca (19.39 mg kg$^{-1}$ FW), Mg (2.80 mg kg$^{-1}$ FW), Zn (25.32 mg kg$^{-1}$ FW) and Fe (0.15 mg kg$^{-1}$ FW) are recorded for Zincol-16 under well-watered conditions (100% FC) and SP with 120 ppm ZnO NPs, followed by Zincol-16 and Ujala-16 when seeds are primed with ZnO NPs at 160 and 120 ppm, respectively. In wheat seeds treated with a high concentration of ZnO NPs (160 ppm) the concentration of P, K, Ca, Mg, Zn and Fe decrease when compared with 120 ppm of ZnO NPs. Moreover, results reveal that Zincol-16 is found to be more responsive to SP of ZnO as compared with Ujala-16.

**Table 2.** Effect of ZnO NPs on root fresh and dry weight, leaf area and leaf water potential of wheat plants.

| Wheat Varieties | Drought Stress | ZnO Levels | Root Fresh Weight (g) | Root Dry Weight (g) | Leaf Area per Plant (cm²) | Leaf Water Potential (Bars) |
|---|---|---|---|---|---|---|
| Ujala-16 | 50% FC | Control | 0.24 j | 0.010 d | 146 j | −77.66 b |
| | | 40 ppm | 0.35 h | 0.020 c | 215 h | −72.00 c |
| | | 80 ppm | 0.42 g | 0.020 c | 254 g | −68.33 d |
| | | 120 ppm | 0.52 e | 0.020 c | 318 e | −62.33 g |
| | | 160 ppm | 0.47 f | 0.020 c | 289 f | −46.66 ij |
| | 100% FC | Control | 0.40 g | 0.020 c | 245 g | −63.33 fg |
| | | 40 ppm | 0.53 de | 0.023 bc | 323 de | −57.00 h |
| | | 80 ppm | 0.54 de | 0.026 ab | 329 de | −48.66 i |
| | | 120 ppm | 0.62 b | 0.030 a | 380 b | −40.00 k |
| | | 160 ppm | 0.58 c | 0.030 a | 355 c | −24.00 m |
| Zincole-16 | 50% FC | Control | 0.29 i | 0.010 d | 180 i | −86.33 a |
| | | 40 ppm | 0.40 g | 0.020 c | 243 g | −79.00 b |
| | | 80 ppm | 0.46 f | 0.020 c | 283 f | −71.66 c |
| | | 120 ppm | 0.55 d | 0.030 a | 335 d | −63.66 fg |
| | | 160 ppm | 0.53 de | 0.026 ab | 326 de | −55.00 h |
| | 100% FC | Control | 0.46 f | 0.020 c | 282 f | −67.33 de |
| | | 40 ppm | 0.52 e | 0.023 bc | 319 e | −65.33 ef |
| | | 80 ppm | 0.58 c | 0.030 a | 354 c | −55.33 h |
| | | 120 ppm | 0.69 a | 0.030 a | 417 a | −46.00 j |
| | | 160 ppm | 0.63 b | 0.030 a | 386 b | −31.66 l |
| LSD value at 5% probability level | | | 0.023 | 0.004 | 13.60 | 2.48 |

Fc = Field capacity. Values having dissimilar letters within same column indicate significant difference at 5% probability.

**Table 3.** Effect of ZnO NPs on relative water content, chlorophyll *a*, *b* and total chlorophyll of wheat plants.

| Wheat Varieties | Drought Stress | ZnO Levels | Relative Water Content (%) | Chlorophyll *a* (mg/g FW) | Chlorophyll *b* (mg/g FW) | Total Chlorophyll (mg/g FW) |
|---|---|---|---|---|---|---|
| Ujala-16 | 50% FC | Control | 63.66 g | 1.35 g | 0.44 f | 1.79 i |
| | | 40 ppm | 64.00 g | 1.36 g | 0.55 e | 1.91 h |
| | | 80 ppm | 77.66 e | 1.65 e | 0.66 c | 2.31 d |
| | | 120 ppm | 79.66 c | 1.69 c | 0.68 b | 2.37 c |
| | | 160 ppm | 79.66 c | 1.69 c | 0.68 b | 2.37 c |
| | 100% FC | Control | 78.66 d | 1.67 d | 0.57 d | 2.24 f |
| | | 40 ppm | 79.66 c | 1.69 c | 0.68 b | 2.37 c |
| | | 80 ppm | 81.66 b | 1.73 b | 0.70 ab | 2.43 b |
| | | 120 ppm | 81.66 b | 1.73 b | 0.70 ab | 2.43 b |
| | | 160 ppm | 82.00 ab | 1.74 ab | 0.71 a | 2.45 ab |

**Table 3.** *Cont.*

| Wheat Varieties | Drought Stress | ZnO Levels | Relative Water Content (%) | Chlorophyll *a* (mg/g FW) | Chlorophyll *b* (mg/g FW) | Total Chlorophyll (mg/g FW) |
|---|---|---|---|---|---|---|
| Zincole-16 | 50% FC | Control | 62.66 h | 1.33 h | 0.43 f | 1.76 j |
| | | 40 ppm | 68.66 f | 1.46 f | 0.59 d | 2.04 g |
| | | 80 ppm | 78.66 d | 1.67 d | 0.66 c | 2.33 d |
| | | 120 ppm | 78.66 d | 1.67 d | 0.65 c | 2.32 d |
| | | 160 ppm | 78.66 d | 1.67 d | 0.66 c | 2.33 d |
| | 100% FC | Control | 79.66 c | 1.69 c | 0.58 d | 2.27 e |
| | | 40 ppm | 79.66 c | 1.69 c | 0.68 b | 2.37 c |
| | | 80 ppm | 79.66 c | 1.69 c | 0.68 b | 2.37 c |
| | | 120 ppm | 82.66 a | 1.75 a | 0.71 a | 2.46 a |
| | | 160 ppm | 79.66 c | 1.69 c | 0.68 b | 2.37 c |
| LSD value at 5% probability level | | | 0.74 | 0.014 | 0.019 | 0.029 |

Fc = Field capacity. Values having dissimilar letters within same column indicate significant difference at 5% probability.

**Table 4.** Effect of ZnO NPs on proline content and antioxidant activity of wheat plants.

| Wheat Varieties | Drought Stress | ZnO Levels | Proline (μmoles $g^{-1}$ FW) | CAT (Units $m^{-1}$ $g^{-1}$ FW) | SOD (Units $m^{-1}$ $g^{-1}$ FW) | APX (ABA Digested $g^{-1}$ FW $h^{-1}$) |
|---|---|---|---|---|---|---|
| Ujala-16 | 50% FC | Control | 1.80 j | 446.68 a | 617.08 a | 2.68 a |
| | | 40 ppm | 3.98 de | 419.95 b | 580.15 b | 2.52 b |
| | | 80 ppm | 4.05 de | 382.47 c | 528.38 c | 2.29 c |
| | | 120 ppm | 4.68 b | 356.74 d | 492.83 d | 2.14 d |
| | | 160 ppm | 4.37 c | 197.59 n | 272.97 n | 1.18 n |
| | 100% FC | Control | 2.01 1 | 217.43 m | 300.37 m | 1.30 m |
| | | 40 ppm | 2.64 h | 229.32 l | 316.80 l | 1.37 l |
| | | 80 ppm | 3.13 g | 278.32 i | 384.50 i | 1.67 i |
| | | 120 ppm | 3.91 e | 301.31 g | 416.26 g | 1.81 g |
| | | 160 ppm | 3.55 f | 269.96 jk | 372.95 jk | 1.62 jk |
| Zincole-16 | 50% FC | Control | 2.21 i | 444.71 a | 614.36 a | 2.67 a |
| | | 40 ppm | 3.92 e | 418.93 b | 578.74 b | 2.51 b |
| | | 80 ppm | 4.36 c | 383.77 c | 530.17 c | 2.30 c |
| | | 120 ppm | 5.13 a | 346.81 e | 479.11 e | 2.08 e |
| | | 160 ppm | 4.74 b | 289.30 h | 399.66 h | 1.73 h |
| | 100% FC | Control | 3.47 f | 212.43 m | 293.47 m | 1.27 m |
| | | 40 ppm | 2.99 g | 234.09 l | 323.39 l | 1.40 l |
| | | 80 ppm | 3.48 f | 268.57 k | 371.03 k | 1.61 k |
| | | 120 ppm | 4.13 d | 309.89 f | 428.12 f | 1.86 f |
| | | 160 ppm | 4.01 de | 276.56 ij | 382.06 ij | 1.66 ij |
| LSD value at 5% probability level | | | 0.167 | 7.10 | 9.81 | 0.042 |

Fc = Field capacity. Values having dissimilar letters within same column indicate significant difference at 5% probability.

**Table 5.** Effect of ZnO NPs on glutathione peroxidase activity, nitrogen, phosphorus and potassium concentration in wheat plants.

| Wheat Varieties | Drought Stress | ZnO Levels | Glutathione Peroxidase (Units $m^{-1}$ $g^{-1}$ FW) | Nitrogen (g $kg^{-1}$ FW) | Phosphorus (g $kg^{-1}$ FW) | Potassium (g $kg^{-1}$ FW) |
|---|---|---|---|---|---|---|
| Ujala-16 | 50% FC | Control | 165.66 a | 2.63 j | 1.75 j | 5.73 j |
| | | 40 ppm | 155.74 b | 3.86 h | 2.03 h | 8.41 h |
| | | 80 ppm | 141.85 c | 4.56 g | 2.76 g | 9.93 g |
| | | 120 ppm | 132.30 d | 5.71 e | 3.96 e | 12.42 e |
| | | 160 ppm | 73.28 n | 5.19 f | 3.41 f | 11.29 f |
| | 100% FC | Control | 80.64 m | 4.40 g | 2.59 g | 9.58 g |
| | | 40 ppm | 85.05 l | 5.82 de | 4.07 de | 12.65 de |
| | | 80 ppm | 103.22 i | 5.91 de | 4.17 de | 12.86 de |
| | | 120 ppm | 111.75 g | 6.84 b | 5.13 b | 14.87 b |
| | | 160 ppm | 100.12 jk | 6.38 c | 4.66 c | 13.88 c |
| Zincole-16 | 50% FC | Control | 164.93 a | 3.23 i | 1.37 i | 7.03 i |
| | | 40 ppm | 155.37 b | 4.37 g | 2.56 g | 9.50 g |
| | | 80 ppm | 142.33 c | 5.09 f | 3.31 f | 11.06 f |
| | | 120 ppm | 128.62 e | 6.03 d | 4.29 d | 13.11 d |
| | | 160 ppm | 107.29 h | 5.86 de | 4.11 de | 12.74 de |
| | 100% FC | Control | 78.78 m | 5.07 f | 3.29 f | 11.03 f |
| | | 40 ppm | 86.81 l | 5.73 e | 3.97 e | 12.45 e |
| | | 80 ppm | 99.60 k | 6.37 c | 4.64 c | 13.85 c |
| | | 120 ppm | 114.93 f | 7.50 a | 5.82 a | 16.31 a |
| | | 160 ppm | 102.57 ij | 6.93 b | 5.22 b | 15.07 b |
| LSD value at 5% probability level | | | 2.63 | 0.24 | 0.25 | 0.53 |

Fc = Field capacity. Values having dissimilar letters within same column indicate significant difference at 5% probability.

**Table 6.** Effect of ZnO NPs on calcium, magnesium, iron and zinc concentration in wheat plants.

| Wheat Varieties | Drought Stress | ZnO Levels | Ca (mg $kg^{-1}$ FW) | Mg (mg $kg^{-1}$ FW) | Fe (mg $kg^{-1}$ FW) | Zn (mg $kg^{-1}$ FW) |
|---|---|---|---|---|---|---|
| Ujala-16 | 50% FC | Control | 6.81 j | 0.98 j | 0.21 a | 8.90 j |
| | | 40 ppm | 9.99 h | 1.44 h | 0.20 b | 13.05 h |
| | | 80 ppm | 11.80 g | 1.70 g | 0.18 c | 15.42 g |
| | | 120 ppm | 14.75 e | 2.13 e | 0.17 d | 19.27 e |
| | | 160 ppm | 13.41 f | 1.93 f | 0.09 k | 17.52 f |
| | 100% FC | Control | 11.38 g | 1.64 g | 0.10 j | 14.86 g |
| | | 40 ppm | 15.04 de | 2.17 de | 0.11 i | 19.64 de |
| | | 80 ppm | 15.29 de | 2.21 de | 0.13 h | 19.96 de |
| | | 120 ppm | 17.67 b | 2.55 b | 0.14 g | 23.08 b |
| | | 160 ppm | 16.49 c | 2.38 c | 0.13 h | 21.54 c |

**Table 6.** *Cont.*

| Wheat Varieties | Drought Stress | ZnO Levels | Ca (mg kg$^{-1}$ FW) | Mg (mg kg$^{-1}$ FW) | Fe (mg kg$^{-1}$ FW) | Zn (mg kg$^{-1}$ FW) |
|---|---|---|---|---|---|---|
| Zincole-16 | 50% FC | Control | 8.36 i | 1.20 i | 0.21 a | 10.92 i |
| | | 40 ppm | 11.29 g | 1.63 g | 0.20 b | 14.75 g |
| | | 80 ppm | 13.15 f | 1.89 f | 0.18 c | 17.17 f |
| | | 120 ppm | 15.59 d | 2.25 d | 0.16 e | 20.36 d |
| | | 160 ppm | 15.14 de | 2.19 de | 0.13 g | 19.78 de |
| | 100% FC | Control | 13.11 f | 1.89 f | 0.10 j | 17.12 f |
| | | 40 ppm | 14.80 e | 2.14 e | 0.11 i | 19.33 e |
| | | 80 ppm | 16.45 c | 2.37 c | 0.13 h | 21.49 c |
| | | 120 ppm | 19.39 a | 2.80 a | 0.15 f | 25.32 a |
| | | 160 ppm | 17.91 b | 2.59 b | 0.13 h | 23.39 b |
| LSD value at 5% probability level | | | 0.63 | 0.092 | 0.0031 | 0.82 |

Fc = Field capacity. Values having dissimilar letters within same column indicate significant difference at 5% probability.

## 4. Discussion

The results of this research trial remain in line with the postulated hypothesis that drought mitigation through SP of ZnO NPs might be developed as a biologically viable strategy to boost wheat growth and its physiological as well as antioxidant mechanisms. The results show that drought stress (DS) decreases agronomic traits while seed priming (SP) with ZnO NPs enhances wheat growth and development under well-watered conditions (100% FC) and DS (50% FC). The significant change in shoot and root lengths, their fresh and dry weights, and leaf area per plants in response to SP with ZnO NPs can be attributed to their physiological and biochemical roles during germination and vegetative growth of wheat [6]. Similar to our findings, it was inferred that ZnO NPs improve the vegetative growth of crop plants depending on the NP's concentrations, however, the underlying mechanisms still await further in-depth research [26,43]. Likewise, it has been inferred that ZnO NPs, when applied in optimum doses, could be more effective in biosynthesizing various endogenous hormones, which tend to mitigate the adverse effects of DS [21]. Additionally, it has been opined that SP with NPs initiates the biosynthesis of hormones such as gibberellin and auxins, which triggers the root and shoot lengths along with fresh and dry weights of crop plants by off-setting the deleterious effects of water scant conditions [43,44]. Moreover, similar to our findings, SP with ZnO NPs was reported to have synchronized the germination and seedling establishment [26]. It was suggested that the Zn application as the SP agent could improve plant growth in the early stages of development by promoting the biosynthesis of growth hormones.

Besides morphological growth, the result of this research suggest that SP with Zn NPs imparts a significant impact on plant physiology and growth because the seeds absorbed a greater concentration of NPs [45]. The increase in plant shoots and root lengths, their fresh weight, and leaf area in plants grown from ZnO NP-primed seeds are presumably increased chlorophyll contents. Improving the biosynthesis process of catalytic and structural components of various proteins, enzymes and co-factor for various developmental pigments was also ascribed to Zn involvement [46,47]. Our results are in line with the previously reported findings by Popović et al. [48] who documented that the application of Zn increases the plant's fresh and dry weight and height due to an increase in chlorophyll content and nutrient acquisition traits under normal conditions. The untreated seeds gain significantly lower biomass, which might be due to Zn deficiency under normal and DS conditions. Similar findings have also been reported by Ljubičić et al. [49] whereby low

concentration of Zn exhibits a slower rate of growth and reduced seedling vigor compared to chemo-priming with ZnO.

Similar to chlorophyl contents, the DS also negatively affects the leaf water potential and relative water content of wheat cultivars, however, SP with ZnO NPs improves these parameters. This might be due to Zn's role in boosting different physiological processes (stomatal regulation, photosynthesis, water use efficiency, cell membrane stability and osmolyte accumulation). These findings are significant and are in concurrence with those reported by [50]. Similarly, it has been inferred that reduction in crop yield is mainly owing to reduced gas exchange rates, uptake of water and leaf water status in plants exposed to DS [4,51]. Under DS, our results reveal that relative water content (RWC) are significantly reduced, which is in agreement with those of [52,53], who documented that DS decreased the relative water content in maize leaves. However, Zn application increases the RWC significantly. The drastic reduction in leaf water potential might be ascribed to a lower RWC, which might have caused stomata closure [4,48,50]. Additionally, the DS causes a significant reduction in chlorophyll content of wheat plants, which might be due to reduced leaf area, premature leaf senescence, increased leaf temperature and impaired photosynthetic machinery [54,55]. However, SP with ZnO NPs increases the chlorophyll content. That increases water uptake and nutrient uptake with the application of ZnO NPs due to better and improved leaf area [25,56]. The ZnO NPs might increase the physiological performance and photosynthesis process. In concurrence with these findings, it is revealed that ZnO NPs exhibit a vital role in the biosynthesis of chlorophyll by protecting the sulfhydryl group of the chlorophyll [57]. The ZnO NPs increase the chlorophyll content by promoting chloroplast development and play a vital role in repairing photosystem by synthesizing a recycling damaged D1 protein [48,49]. Overall, the change in the biomass of crop plants are in agreement with the findings of Salam et al. [46] who state that the NPs ZnO significantly improves the leaf pigments in plants, which enhances the biomass productivity in stressful environment.

Proline content and the activity of antioxidants such as APX, CAT, GPx and SOD, increases in the plants grown under the DS conditions in comparison with well-watered conditions. However, under the well-watered and DS conditions, the increases in proline and antioxidant content are noted for SP with ZnO NPs. This might be attributed to the biosynthesis of Zn finger proteins. ROS scavenging is enhanced by the $C_2H_2$ Zn finger protein. The $C_2H_2$ Zn finger protein boosts drought tolerance in plants. Scavenging the ROS is owed to increased activities of SOD and POD in rice by the ZFP245 Zn finger protein. These reduce the deleterious impacts of DS and boost the growth and paddy yield [58]. Moreover in plants, the DS increases with the $C_2H_2$ Zn finger protein which imparts drought tolerance in the plant by activating the signaling process and triggering the biosynthesis of ABA hormone [59]. Therefore, the increase in the expression of Zn finger proteins counters the adverse effects of DS by increasing the synthesis of compatible solutes, scavenging ROS and triggering the signaling pathways. In this trial, the DS decreases the uptake of N, P, K, Ca, Mg, Zn and Fe in plants in comparison with well-watered conditions. Moreover, SP with ZnO NPs remains effective in increasing the uptake of N, K, and Zn as compared with P, Ca, Mg and Fe, which might be due to the negative interaction of Mg, Fe, P and Ca in the Zn absorption on the surface of the root and its translocation from root to shoot in plants [44,60].

## 5. Conclusions

This study was performed to evaluate the drought tolerance of wheat varieties through seed priming with ZnO nanoparticles. The results show that the leaf pigments significantly decrease under the DS conditions, while improvements are observed where seed priming with ZnO NPs is applied at the rate of 120 ppm as compared with all other treatments. Similarly, the activity of antioxidants, as well as the nutrient contents of wheat, increase where ZnO nanoparticles are applied and show better improvement under drought as well as normal conditions. Among both varieties, the wheat variety Zincole-16 performs better

as compared with Ujala-16 under normal as well as drought-stress conditions where ZnO nanoparticles are primed. Therefore, ZnO nanoparticle seed priming might be developed as a biologically viable approach for improving the performance of wheat under water-scant conditions.

**Author Contributions:** Conception and design were proposed by S.F.A. and M.A.B. and collected the data. S.F.A., M.A.I. and Z.A. assembled the data and prepared the first draft of the manuscript. M.A.S.R., M.A.I., M.D.A., E.F.A., K.F.A. and G.H.A. finalized the final draft of the manuscript. This article was funded by M.D.A., E.F.A. and K.F.A. M.A.I. improved and English Editing. All authors have read and agreed to the published version of the manuscript.

**Funding:** The authors extend their appreciation to Princess Nourah bint Abdulrahman University Researchers Supporting Project number (PNURSP2023R355), Princess Nourah bint Abdulrahman University, Riyadh, Saudi Arabia.

**Institutional Review Board Statement:** Not applicable.

**Informed Consent Statement:** Not applicable.

**Data Availability Statement:** Most of the recorded data are available in all tables in the manuscript.

**Acknowledgments:** The authors extend their appreciation to Princess Nourah bint Abdulrahman University Researchers Supporting Project number (PNURSP2023R355), Princess Nourah bint Abdulrahman University, Riyadh, Saudi Arabia. The authors gratefully acknowledge technical support provided by the Stress Physiology Lab, Department of Agronomy, The Islamia University of Bahawalpur for sharing lab facilities and providing space for research experiments.

**Conflicts of Interest:** The authors declare no conflict of interest.

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
