# Peer review of "Enhancing Drought Tolerance in Wheat Cultivars through Nano-ZnO Priming by Improving Leaf Pigments and Antioxidant Activity"

_sustainability, doi:10.3390/su15075835_

Round 1

Reviewer 1 Report

The present manuscript entitled “Enhancing drought tolerance in wheat cultivars through nano-ZnO priming by improving leaf pigments and antioxidant activity” by Abbas et al., describes the two wheat cultivars (Ujala-16 and Zincol-16) that were subjected to pre-sowing priming treatments with different doses of ZnO NPs (40, 80, 120 and 160 ppm) under 50% and 100% field capacity (FC) conditions. Furthermore, the results exhibited that Zincol-16 remained superior to Ujala-16, while ZnO NPs increased the growth and development of both wheat varieties and thus this combination might be recommended to wheat growers after testing further in-depth evaluation of more doses of ZnO NPs. The authors report an interesting work. The objective and justification of the work are clear. However, I recommend it for publication after certain Minor corrections are detailed below which need to be addressed before its final acceptance in Sustainability.

I advise the authors to take the following points into account while revising their manuscript.

Comment 1: There are so many typographical, superscript, and subscript errors in the manuscript text, so the authors need to correct them in the revised manuscript.

Comment 2: English needs to be a little improved, as there are some misused conjunctions and technical flaws. So it needs to be corrected in the manuscript.

Comment 3: The abstract does not clearly state the findings of this study and also needs to include the details of the preparation of ZnO NPs and performed characterization techniques. So the abstract section needs to be rewritten.

Comment 4: Include the details of the used instruments in characterization section 2, such as model number, manufacturer, and place of origin, etc.,

Comment 5: The authors need to include the details of the preparation of ZnO NPs in the revised manuscript text.

Comment 6: Authors mentioned that the characterization of ZnO NPs was done by using UV–visible spectroscopy, transmission electron microscopy (TEM), and X-Ray diffraction (XRD). However, they did not provide any details of the characterization details and the results. So the authors must need to include the TEM, UV-Vis, and XRD results of ZnO NPs in the revised manuscript text.

Comment 7: Include the structured graphical abstract in the revised manuscript.

Comment 8: Include the equation numbers e.g. (1), (2) & (3), etc., to the mentioned equations in section 2.2.

Comment 9: Revise and elaborate the Conclusion section with important findings

Comment 10: The homogeneity of the reference section needs to be maintained. In some references, journal names are written in full form and some in abbreviation form. So please check and revise accordingly to the journal's instructions.

Author Response

Respected Reviewer

Hope you will be fine with good health.

All your good suggestion and comments are added in revised manuscript.

Please see in attachment of your response report.

Thanks and Regards

Dr. Zahoor Ahmad

Reviewer 2 Report

Results of present study have scientific sound. Overall, quality of manuscript is fine. However, I have some suggestions-

1. Antioxidant enzymes activities such as peroxidase (POX), catalase (CAT), ascorbate  peroxidase (APX), and glutathione peroxidase dont have methodology. For audience purpose, methodology of above antioxidant should be described properly.

2. I want to suggest for improving discussion part by adding some new references for interpretation of results clearly. 

Author Response

(The authors gave the same response as above.)

Reviewer 3 Report

Dear Authors,

thank you for the opportunity to meet the manuscript entitled: "Enhancing drought tolerance in wheat cultivars through nano-ZnO priming by improving leaf pigments and antioxidant activity". Currently, the topic of nanoparticles application in agriculture is very widespread, so I evaluate the choice of topic very positively. I also positively evaluate the number of observed traits on two different wheat varieties.

However, I have several comments regarding the preparation of the manuscript.

Formal and linguistic editing of the text across the entire manuscript is needed.

My most important comment is connected with the Results chapter. Despite the fact that in most cases the evaluation follows the results in the tables, the authors missed the opportunity to evaluate the essential findings in the context of the topic of the experiment.

The topic of the manuscript was the effect of ZnO priming seeds on the performance of wheat in dry conditions. However, in most cases, the authors focused on evaluating the results in 100% FC, because in these conditions the results were generally the best. However, this is expected and therefore I think that much more space should have been devoted to results and comparisons in dry conditions.

Other comments are marked in the attachment.

Author Response

(The authors gave the same response as above.)

Reviewer 4 Report

The presented study investigates the impact of treating wheat crop seeds with different doses of ZnO nanoparticles on crop capacity to tolerate drought.

The study is interesting, especially under the ongoing concern about climate change and its potential 'negative' impact that might affect crop production and food security. Authors considered some crop physiological and biocemical parameters and nutrient content in response of the applied ZnO nanoparticle doses. However, I would reccommend and encourage authors to also test the impact of applied treatments on water use efficiency, grain yield, and water productivity. It will help the reader to understand the potential impact on water savings especially in arid and semi-arid regions, and the potential yield; that contributes directly or indirectly in food security and farmer net revenue. 

Further minor comments are included in the attached file and highlighted in yellow.

Author Response

(The authors gave the same response as above.)

Reviewer 5 Report

1. The title is in accordance with the study carried out.

 2. The abstract is concise.

3. Material and research methods described are in accordance with the study. 

However, are needed minor corrections to improve the quality of the manuscript:

Regarding the material and method - I recommend specifying the duration of the experiment from sowing to the moment when you started to analyze the plants subjected to the treatments

I recommend inserting some photos during the experiment.

L125 .... deleting the number 0...assuming that you wanted to mention the number of plants analyzed (5)

L138 - RWC = [(FW-DW) / (RW-DW)] x 100 – the formula is wrong, you must change RW with TW

L 141, 148, 149, 150, 155, 163, 164, 168, 170 standardizations of the unit of measure ml or mL

L 142 – I recommend the correct indication of the author, in the quoted source "[31]" the equations regarding the determination of chlorophyll are not found. The chlorophyll a, band total Chl contents were determined using the following equations [31]

L147 - correction of estmation in estimation

L151 - erasing the space between the number and degrees 100 0C

L165 - temperature change 350C in 350C

L170 – correction Therafter – Thereafter

L174  correction estmates in estimates

L190 correction andwater-stressed in and water

L232 correction antioxidant in antioxidants

L253 – correction wth in with

L257 – correction incrementn in increment

L289 – correction scientist in scientists

L314 – inserting a space ofDS

Correctly cite bibliographic source number 23...specifying the Name

23. A, S.; A, S.; N, M. Role of Zinc Nutrition for Increasing Zinc Availability, Uptake, Yield, and Quality of Maize (Zea Mays 413 L.) Grains: An Overview. Commun. Soil Sci. Plant Anal. 2020, 51, 2001–2021.

Sugestion

Suganya A., Saravanan A. & Manivannan N (2020) Role of Zinc Nutrition for Increasing Zinc Availability, Uptake, Yield, and Quality of Maize (Zea Mays L.) Grains: An Overview, Communications in Soil Science and Plant Analysis, 51:15, 2001-2021, DOI: 10.1080/00103624.2020.1820030

Space between genus and species at bibliographic source 43

43. Munir, T.; Rizwan, M.; Kashif, M.; Shahzad, A.; Ali, S.; Amin, N.; Zahid, R.; Alam, M.F.E.; Imran, M. Effect of Zinc 469 Oxide Nanoparticles on the Growth and Zn Uptake in Wheat (Triticumaestivum L.) by Seed Priming Method. Dig. J. Na- 470 nomater. Biostructures 2018, 13, 315–323.

Author Response

(The authors gave the same response as above.)

Round 2

Reviewer 2 Report

Authors have improved quality of manuscript. Now looking nice and find suitable for acceptance. 

Reviewer 3 Report

Thank you for taking my comments into account and editing the manuscript based on them. I believe it could have increased its quality.